# Optimizing respiratory virus surveillance networks using uncertainty propagation

Sen Pei [1✉], Xian Teng[2], Paul Lewis[3] & Jeffrey Shaman [1✉]

Infectious disease prevention, control and forecasting rely on sentinel observations; however, many locations lack the capacity for routine surveillance. Here we show that, by using data from multiple sites collectively, accurate estimation and forecasting of respiratory diseases for locations without surveillance is feasible. We develop a framework to optimize surveillance sites that suppresses uncertainty propagation in a networked disease transmission model. Using influenza outbreaks from 35 US states, the optimized system generates better near-term predictions than alternate systems designed using population and human mobility. We also find that monitoring regional population centers serves as a reasonable proxy for the optimized network and could direct surveillance for diseases with limited records. The proxy method is validated using model simulations for 3,108 US counties and historical data for two other respiratory pathogens – human metapneumovirus and seasonal coronavirus – from 35 US states and can be used to guide systemic allocation of surveillance efforts.

[1] Department of Environmental Health Sciences, Mailman School of Public Health, Columbia University, New York, NY 10032, USA. [2] School of Computing and Information, University of Pittsburgh, Pittsburgh, PA 15260, USA. [3] Integrated Biosurveillance Section, Armed Forces Health Surveillance Branch, Silver Spring, MD 20904, USA. ✉email: sp3449@cumc.columbia.edu; jls106@cumc.columbia.edu

Respiratory viruses impose a high morbidity and mortality burden on human health globally: influenza alone claims 290,000 to 650,000 lives worldwide each year[1]. Sentinel surveillance and operational real-time forecasting systems are decision support tools that help improve the prevention and control of these pathogens[2]. A number of forecasting methods for influenza have been developed recently[3–14]. In the last few years, some of these systems have been applied operationally to forecast influenza outbreaks in the United States[15–17], demonstrating the feasibility of real-time prediction.

Surveillance data are necessary to support real-time operational forecasting. However, many locations lack sufficient resources to maintain high-quality, continuous surveillance[18–20]. This data shortcoming limits infectious disease monitoring and forecasting at those sites. At the same time, network modeling approaches that dynamically couple disease transmission across multiple locations are widely used for infectious disease simulation[21–24]. These models have been recently leveraged to simulate, monitor, and forecast epidemic outbreaks. For instance, metapopulation models informed by observed human movement (air-transportation[25–27], mobile phone location[28,29], work commuting[30–32], etc.) have supported better understanding and forecasting of the spatial spread of influenza[13,26,27,33,34], dengue[29], malaria[28], and COVID-19[35–38]. Further, statistical correlations of disease activity at multiple sites have enabled improved surveillance of real-time influenza incidence (i.e., nowcasting)[39]. This coupling of disease activity through time and across locations suggests that infectious disease monitoring and forecasting at locations lacking surveillance capacity may be possible. To support such efforts, there is a need for developing methods that optimize disease surveillance and forecasting using incomplete data.

A number of studies have explored the optimization of disease surveillance systems from a variety of perspectives. Approaches include the development of a method to select sentinel providers for influenza in Iowa that maximizes the population covered by the surveillance network[18] and the design of surveillance systems that sequentially recruit sentinel sites that most improve system estimation of influenza-like illness hospitalizations[19]. This latter optimization method, applied to influenza surveillance in Texas[19] and arbovirus surveillance in Puerto Rico[40], employs submodular optimization to provide a performance guarantee[41]. Another approach evaluated strategies for selecting sensors in a social network and found that the optimal choice depends on public health goals, network structure, and disease transmissibility[42]. More recently, there has been a growing interest in combining and optimizing the inclusion of non-traditional data sources such as online search queries and social media activities[43,44].

In this study, we demonstrate that forecasting for locations without surveillance is possible using data streams from multiple other locations collectively in a networked, mechanistic, forecasting system informed by human movement (see "Materials and Methods"). In this system, a mobility-driven metapopulation model describing the spatiotemporal transmission of respiratory virus across locations is iteratively updated using the latest observed incidence[13]. Observations from one location are used to adjust the model state and estimate incidence in other locations, including those without surveillance. The optimized model is then evolved into the future to generate forecasts (Fig. 1a). Such networked systems enable inference and prediction of local disease activity in locations lacking observations and provide a framework for designing cost-effective surveillance and forecasting systems in circumstances constrained by limited resources.

## Results

**Forecasting with incomplete information**. We performed a preliminary forecasting experiment for influenza outbreaks in 35 US states in which data from a single surveillance site were omitted. Specifically, we used the ILI (influenza-like illness) rate among all people seeking medical attention multiplied by the percentage of patients with laboratory-confirmed influenza type A, termed ILI+[45], to estimate local influenza activity (Fig. 1b, Methods, Supplementary Note 1 and Supplementary Fig. 1). In the experiment, a set of forecasts was generated with data inputs from all 35 states over 9 seasons, and a second set of forecasts over 9 seasons was generated with data inputs from 34 states, omitting data from one state in turn (Supplementary Note 2). The forecast mean absolute error in the omitted state was averaged over all 35 locations for versions with and without surveillance data. Forecast errors of near-term predictions for 1- to 4-week ahead ILI+ indicate that omitting data from a single surveillance site does not seriously degrade forecast accuracy in the omitted locations (Fig. 1c and Supplementary Fig. 2).

The ability to estimate and forecast disease activity for locations without observations poses an additional question: can a limited number of surveillance sites be optimally identified in order to support accurate estimation and forecasting of disease activity at all sites in a network? This question motivates the design of a quantitative framework for the optimal selection of surveillance sites within a network. In disease surveillance, incomplete and imperfect observation leads to uncertainty in the estimation of disease activity, which disrupts surveillance, forecasting, and prevention and control efforts. This uncertainty should be minimized (see discussions in Supplementary Note 3 and Supplementary Fig. 3); however, due to the nonlinear evolution of infectious disease transmission, uncertainty can grow over time[46,47]. This uncertainty propagation compromises the accuracy of both surveillance and forecasting: accumulated uncertainty growth from prior observations can undermine the understanding of the current disease situation (i.e., surveillance), and prospective uncertainty growth can limit prediction of future incidence (i.e., forecast). This effect is clearly evident in influenza forecasting for which smaller uncertainty across a forecast ensemble generally implies a better prediction[3,5,10,13]. Leveraging this relationship, an effective surveillance network should be designed to collect the most informative data that best suppresses uncertainty growth.

**Uncertainty propagation**. Here we develop a framework to quantify the spatiotemporal propagation of uncertainty in a networked forecasting system. We characterize the evolution of uncertainty in the estimated infected and susceptible populations. For $m$ locations, a binary vector $\mathbf{p} = (p_1,\ldots, p_m)^{\mathrm{T}}$ is used to record whether location $i$ is selected for surveillance ($p_i = 1$) or omitted ($p_i = 0$). We denote the vector of uncertainty as $\mathbf{x} = \left(\sigma_{I_1}, \ldots, \sigma_{I_m}, \sigma_{S_1}, \ldots, \sigma_{S_m}\right)^{\mathrm{T}}$, where $\sigma_{I_i}$ and $\sigma_{S_i}$ represent the uncertainty (here measured by standard deviation) in the estimated infected and susceptible populations at location $i$. The propagation of $\mathbf{x}$ undergoes two interacting processes during the generation of a forecast: uncertainty reduction during model update using data assimilation methods and uncertainty growth during model integration (Fig. 1d). The evolution of the uncertainty vector during a short time interval can be approximated using a linear operation: $\mathbf{x} \rightarrow \mathbf{MPx}$, where the diagonal matrix $\mathbf{P}$ quantifies the uncertainty reduction during data assimilation, and the matrix $\mathbf{M}$ estimates uncertainty growth in the dynamical model.

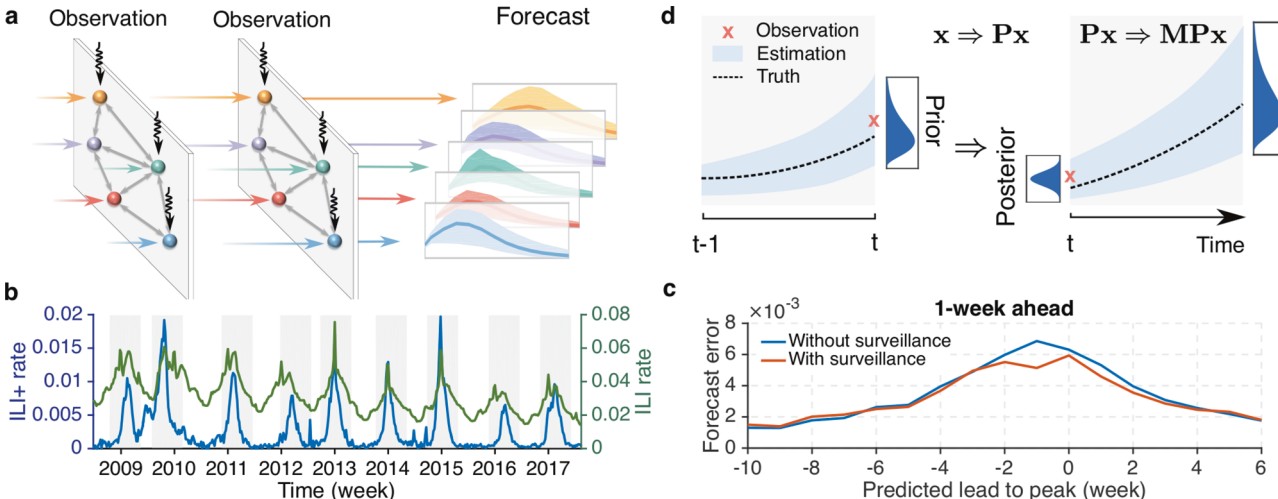

**Fig. 1 The networked forecasting system and uncertainty propagation. a** Schematic illustration of the networked forecasting system. At each observation time point, incidence data from 3 locations (vertical arrows) are used to adjust the dynamical model consisting of 5 connected locations. The adjusted model is then evolved forward (horizontal arrows) to the next observation time point, and ultimately further into the future to generate a forecast. (**b**) The national ILI+ rate (blue line) and ILI rate (green line) for the 2008–2009 to 2016–2017 seasons from AFHSB data. Shaded areas indicate the retrospective forecasting periods. (**c**) Comparison of forecast error (mean absolute error) for 1-week ahead prediction with (red line) and without (blue line) surveillance data. The forecast error at each predicted lead (negative/positive: before/after predicted peak) was averaged over all 35 locations for versions with and without surveillance data. (**d**) Uncertainty propagation in the networked forecasting system. At time $t$, the prior state is updated to a posterior using available observations (red cross), which constrains the model toward the truth (dash line). The reduction of uncertainty $\mathbf{x}$ due to data assimilation and its growth during model integration can be approximated by $\mathbf{x} \Rightarrow \mathbf{Px}$ and $\mathbf{Px} \Rightarrow \mathbf{MPx}$.

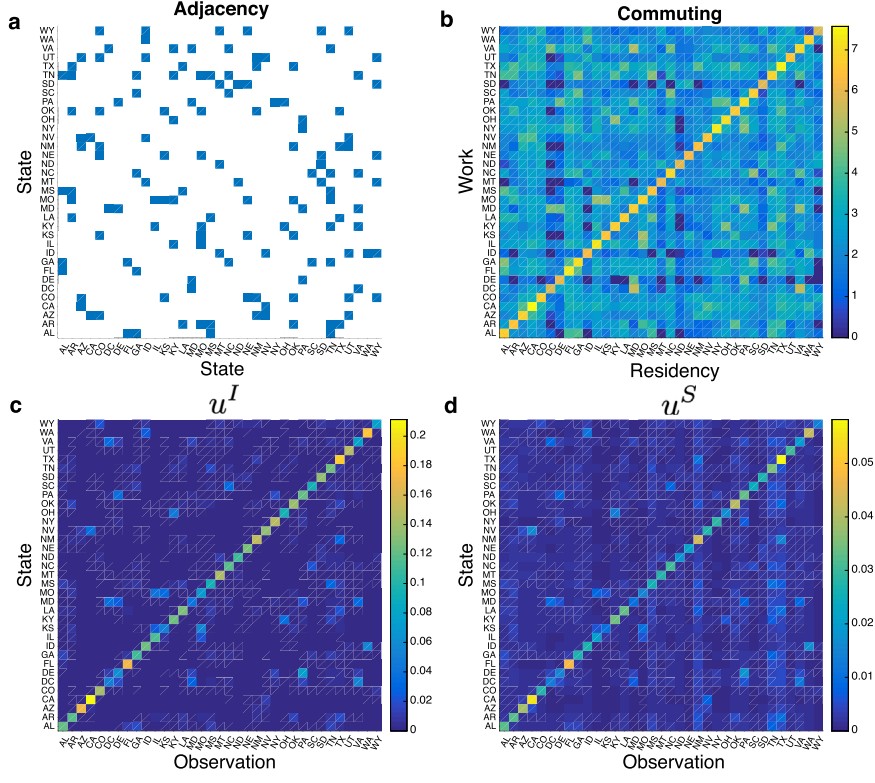

**Fig. 2 Connectivity and uncertainty reduction across 35 US states. a** The adjacency matrix for 35 US states. Adjacent states are highlighted by blue squares. **b** Numbers of commuters among 35 US states from the 2010 census survey. Color shows the logarithmic-transformed (base 10) commuting population from resident (x-axis) to work (y-axis) locations. Surveillance data from one state (x-axis) can reduce the uncertainty of infected (**c**) and susceptible (**d**) populations in other states (y-axis). Color indicates the reduced fraction of variance for infected and susceptible populations ($u^I$ and $u^S$). Results are averaged over data assimilation during nine seasons.

Disease transmission dynamics in different locations are coupled in the mobility-driven metapopulation model. The adjacency matrix and numbers of commuters among the examined 35 US states are presented in Fig. 2a–b. The dynamical coupling enables the adjustment of infected and susceptible populations in one location using surveillance data from another. To quantify uncertainty reduction during this adjustment, we introduce a diagonal matrix $\mathbf{P} = \text{diag}\,(P_1,\ldots,P_m,P_{m+1},\ldots,P_{2m})$ with the diagonal elements defined as

$$P_j = \sqrt{\prod_{i=1}^{m}\left(1 - p_i u_{j\leftarrow i}^I\right)}, \quad P_{j+m} = \sqrt{\prod_{i=1}^{m}\left(1 - p_i u_{j\leftarrow i}^S\right)} \quad (1)$$

for $j = 1,\ldots,m$. Here, $u_{j\leftarrow i}^I$ and $u_{j\leftarrow i}^S$ are the fractional *variance* reduction for the infected and susceptible populations in location $j$ attributed to the observation from location $i$. The matrix $\mathbf{P}$ encodes information about the surveillance network configuration $\mathbf{p}$: if a location has observations (i.e., $p_i = 1$), uncertainty in this location and other dynamically coupled locations is reduced; otherwise (i.e., $p_i = 0$), this location makes no contribution to uncertainty reduction. After data assimilation, the prior model state is adjusted to a posterior, with the uncertain vector $\mathbf{x}$ updated to $\mathbf{Px}$. The surveillance network configuration $\mathbf{p}$ determines the diagonal elements of $\mathbf{P}$, thus controls the reduction of the uncertainty vector $\mathbf{x}$.

The values of $u_{j\leftarrow i}^I$ and $u_{j\leftarrow i}^S$ depend on the quality of the observation in location $i$. Particularly, surveillance data with less uncertainty, characterized by a smaller observational error variance (OEV), lead to a larger reduction of uncertainty in $\mathbf{x}$. Thus, to calculate $u_{j\leftarrow i}^I$ and $u_{j\leftarrow i}^S$, a precise estimation of OEV is required; however, in practice, this is a challenging task as only one data point (ILI+) is observed per location per week. We therefore developed a method to quantify the OEV of these observations and reveal that the OEV of ILI+ is predominantly affected by the number of laboratory tests (Supplementary Note 4). In order to properly represent the uncertainty of observations, we optimized the OEV of ILI+ from different locations in retrospective forecasting so that near-term forecast error is minimized (Supplementary Fig. 4). The forms of cross-location uncertainty reduction $u_{j\leftarrow i}^I$ and $u_{j\leftarrow i}^S$ are derived using a state-space framework (Supplementary Note 5 and Supplementary Fig 5) and reported in Methods.

We computed the mean values of $u_{j\leftarrow i}^I$ and $u_{j\leftarrow i}^S$ averaged over weekly influenza forecasts during 9 seasons. The surveillance data from one location $i$ mostly affect the uncertainty of its own infected and susceptible populations (Fig. 2c–d, diagonal elements); however, for certain locations that are adjacent to location $i$ or exchange a large number of commuters (Fig. 2a–b), the variances of infected and susceptible populations are reduced by the observation from location $i$ as well (Fig. 2c–d, off-diagonal elements). Such cross-site uncertainty reduction indicates dynamical coupling between these pairs of locations.

The reduced uncertainty $\mathbf{Px}$ will propagate in the networked system during model integration. The evolution of $\mathbf{Px}$ within a short time interval can be approximated using the linear propagator $\mathbf{M}$ of the transmission model that characterizes the uncertainty growth driven by the linearized model dynamics: $\mathbf{Px} \rightarrow \mathbf{MPx}$. Specifically, for a short time interval $\delta t$, the linear propagator $\mathbf{M}$ is estimated by $\mathbf{M} \approx \mathbf{I} + \mathbf{J}\delta t$, where $\mathbf{I}$ is a $2m \times 2m$ unit matrix and $\mathbf{J}$ is the Jacobian matrix of the full nonlinear system (Supplementary Note 5). The linear approximation was shown to be valid for a few days for influenza transmission models[47], and has been previously applied in numerical weather prediction[46]. Typical respiratory disease surveillance releases data once per week[2]; at this rate the linear approximation may become less accurate. As a consequence, we here limit our attention to

short-term uncertainty propagation. Later retrospective forecast results indicate that this setting can improve near-term forecasts for ILI+ up to 4 weeks ahead.

**The optimal surveillance problem**. To minimize uncertainty growth during short-term forecast, we aim to minimize the uncertainty growth rate, quantified by $\|\mathbf{MPx}\|/\|\mathbf{x}\| = (\mathbf{x}^T\,\mathbf{P}^T\,\mathbf{M}^T\,\mathbf{MPx})/(\mathbf{x}^T\,\mathbf{x})$[46,47]. This equation indicates that the uncertainty growth rate is determined by the dominant eigenvalue, $\lambda_1$, of the matrix $\mathbf{L} \equiv \mathbf{P}^T\,\mathbf{M}^T\,\mathbf{MP}$. In operation, the matrices $\mathbf{P}$ and $\mathbf{M}$ vary by forecast time (i.e., how far into an outbreak a forecast is initiated) and system state. Thus, to design an optimal surveillance network for a wide range of unknown, potential outbreaks, we minimize the mean value, $\langle\lambda_1\rangle$, averaged over different forecast initiation time points and system states. Mathematically, the task of selecting $K$ optimal sentinel sites from $m$ locations is transformed to the combinatorial optimization problem of finding $\mathbf{p}$ that minimizes $\langle\lambda_1\rangle$ under the constraint $\sum_{i=1}^{m} p_i = K$:

$$\mathbf{p}^* = \arg\min\langle\lambda_1(\mathbf{p},t,\mathbf{z})\rangle\,\text{subject to}\,\sum_{i=1}^{m} p_i = K, p_i \in \{0,1\}. \quad (2)$$

Here $\lambda_1\,(\mathbf{p},\,t,\,\mathbf{z})$ is the dominant eigenvalue of $\mathbf{L}$ at time $t$ with system state $\mathbf{z}$ given the configuration of the surveillance network $\mathbf{p}$. In order to calculate $\lambda_1$, we run weekly data assimilation in multiple seasons to estimate the system state $\mathbf{z}$ at each week. Using the surveillance network configuration $\mathbf{p}$ and the posterior model state $\mathbf{z}$ at time $t$, we obtain the matrices $\mathbf{P}$ and $\mathbf{M}$, and then compute the dominant eigenvalue $\lambda_1$ of $\mathbf{L}$ using the power method[48]. The mean eigenvalue is averaged over $\lambda_1(\mathbf{p},\,t,\,\mathbf{z})$ for different weeks and seasons.

The above optimal surveillance problem is a combinatorial optimization, as the inclusion of one location is impacted by other selected locations. Solving this problem for large-scale systems is challenging as the number of configurations grows exponentially with the system size. However, for a small system forecasting respiratory disease at the US state level, this problem can be solved using standard iterative optimization techniques such as simulated annealing (SA)[49] (Methods).

**Influenza surveillance networks**. We validated the proposed framework using influenza outbreaks in 35 US states. In order to perform the optimization, historical outbreak data are required to infer model parameters and state variables so that simulation dynamics are representative of real-world influenza transmission patterns (e.g., seasonality, spatiotemporal spread, typical attack rate, etc.). Although sentinel providers tend to work locally, in practice, surveillance data collected from local sentinel providers are aggregated to coarser geographical scales for public health use. In particular, the US Centers for Disease Control and Prevention (CDC) releases ILI surveillance data at the state, HHS (the US Department of Health and Human Services) regional and national levels[50]. Here we work at this operational spatial resolution and optimize the surveillance networks at the state level.

For a given number of observation locations, $K$, we optimize the surveillance network using SA. As short-term uncertainty propagation is suppressed, we expect that the forecast accuracy of the selected network for near-term targets, for instance, 1- to 4-week ahead ILI+, will outperform surveillance systems designed using heuristic strategies that favor locations with either larger population size, a larger number of commuters (both incoming and outgoing directions), or a higher population gradient ($\nabla$Population, defined as the ratio of location population size to the average population of its adjacent neighbors).

Among all strategies, SA is best at minimizing the average dominant eigenvalue (Fig. 3a), and the selected states (for $K = 5$,

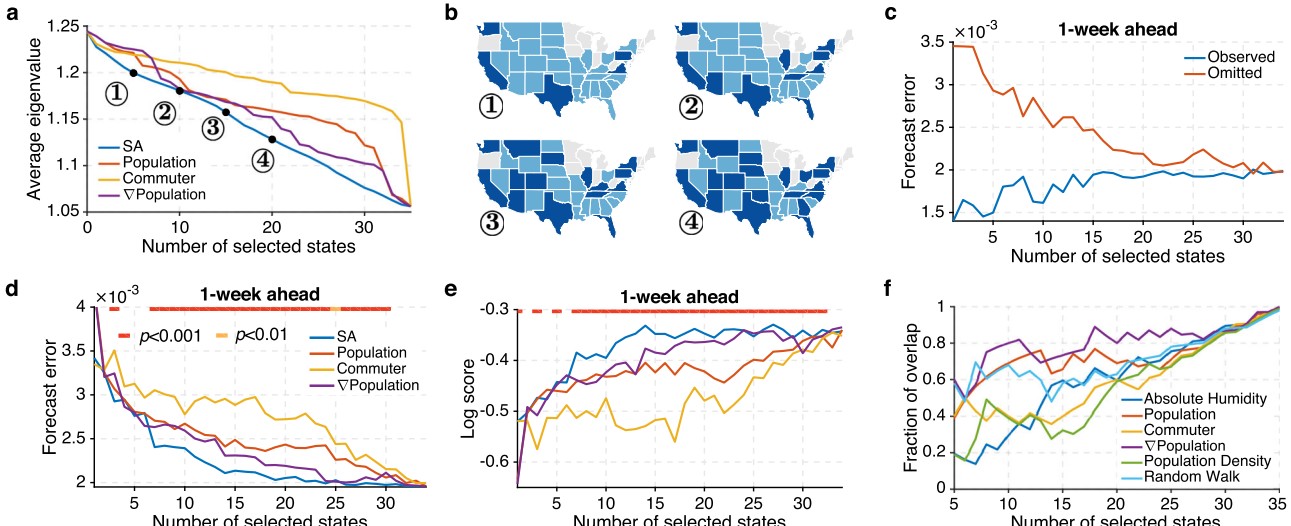

**Fig. 3 Surveillance network optimization for 35 US states. a** The average eigenvalues of surveillance networks selected by SA (simulated annealing) (blue line), population (red line), commuter numbers (orange line), and population gradient (∇Population) (purple line). Four SA surveillance networks are displayed in (**b**). Dark blue states are selected by the SA optimization. Grey states were not included in the analysis. **c** Forecast error for 1-week ahead ILI+ prediction in the observed (blue line) and omitted (red line) states. Surveillance locations are selected by SA. **d** Forecast error for 1-week ahead ILI+ prediction using surveillance networks designed by different methods. The horizontal bar on top indicates the statistical significance for SA outperforming all other methods (two-sided Wilcoxon signed-rank test; red: $p < 0.001$, orange: $p < 0.01$, none: $p \geq 0.01$). **e** Log score for 1-week ahead ILI+ prediction using surveillance networks designed by different methods. **f** Overlap between the states selected by SA and those selected by other attributes: absolute humidity (blue line), population (red line), commuter (orange line), ∇Population (purple line), population density (green line), and random walk centrality (light blue line).

10, 15, 20) are spread across the country (Fig. 3b). We next performed retrospective forecasting for 9 seasons at the state level (Methods, Supplementary Note 6 and Supplementary Fig. 6). In retrospective forecasting, all 35 states were included in the metapopulation model, but only surveillance data from selected states were used to calibrate the model (i.e., observations from unselected states were omitted). Using the surveillance networks optimized by SA, the forecast error of near-term predictions in the states without surveillance decreases as more states are observed, and eventually converges to the forecast error of the states with observations (Fig. 3c).

To evaluate the performance of surveillance networks selected using different methods, we compared the forecast error (mean absolute error) for 1-week ahead ILI+ predictions in all states, including those with and without surveillance data. In most cases, the SA approach significantly outperforms the other heuristic methods by generating surveillance networks that support lower forecast error (Fig. 3d, Wilcoxon signed-rank test, Methods and Supplementary Fig. 7). The marginal gain of observing more locations gradually decreases, highlighting the dominant role that observations from certain key locations play in constraining influenza forecast accuracy. Comparison for 2- to 4-week ahead predictions (Supplementary Fig. 7) additionally corroborate the effective minimization of uncertainty growth by SA optimization.

The forecasting system generates probabilistic forecasts. Mean absolute errors reported in Fig. 3c only measure the error of point prediction (i.e., the mean value of each ensemble forecast). In order to evaluate the full probabilistic forecasts, we compared the "log score" (Methods), defined as the logarithmic value of the probability assigned to an interval around the observed target. In essence, the log score is a summary statistic measuring the distribution of ensemble forecast error. This probabilistic scoring rule has been used in the CDC FluSight forecast challenge[15–17]. Consistent with the results for forecast error, the SA approach outperforms the other three strategies (Fig. 3e). We further examined the forecast error and log score at different times

relative to the predicted peak week (Supplementary Figs. 8–9). As an example, retrospective forecasts were generated for all 35 states over 9 seasons using surveillance networks consisting of 20 states. At most predicted lead weeks, the SA optimization supports better predictions.

To understand the features of networks selected by SA, we examined their similarity with networks identified using alternate heuristic methods. In addition to population size, number of commuters, and population gradient, we also investigated three other feature-driven surveillance location selection methods and compared their results with those selected by SA. These features are: (1) Absolute humidity. In temperate regions, influenza transmission is favored during periods of lower absolute humidity[51]. As a result, we selected locations with lower average absolute humidity with priority. (2) Population density. Higher population density may facilitate influenza transmission due to higher person-to-person contact frequency. Locations are ranked by their population density in descending order. (3) Random walk centrality. In contrast to other local features, random walk centrality is a global metric determined by the connectivity among all locations. Specifically, the random walk centrality $r_i$ for location $i$ is the stationary visiting probability of a random walker who travels in the network following the transfer probability specified by the commuting matrix. The values of $r_i$ satisfy the self-consistent equation: $r_i = \sum_j C_i^j r_j / N_j$, and can be calculated through iteration ($C_i^j$ is the number of commuters from location $j$ to $i$, and $N_j$ is the population in location $j$). For random walk centrality, locations are ranked according to $r_i$ in descending order.

Among all examined measures, the ∇Population approach is most similar to the eigenvalue minimization approach using SA (Fig. 3f), indicating that the optimized network has a tendency to first select locations with a high ∇Population. For example, Washington state ranks only 11th and 25th by population and number of commuters among 35 examined US states; however, it

ranks 3rd according to ∇Population and is selected with high priority by the eigenvalue minimization approach.

An attractive alternative approach to SA to solve the optimal surveillance problem is to sequentially add locations that produce the largest marginal reduction of the eigenvalue. This greedy approach is less computationally demanding than the SA algorithm, and could have a performance guarantee if the objective function satisfies the submodular property[41]. A function is submodular if the marginal gain of including an additional location decreases with the number of existing surveillance sites. Unfortunately, the eigenvalue function we use here does not have this diminishing return property. Despite this circumstance, we tested a greedy algorithm approach and compared the resulting eigenvalue with the one obtained from the SA algorithm (Supplementary Fig. 10). The eigenvalue curves are identical for surveillance systems with less than 15 states and remain similar for larger systems. These findings indicate that the greedy approach is effective for this 35-state model, and may be applicable to small- and medium-sized systems. However, for large systems like the county-level transmission model, the greedy algorithm is computationally prohibitive due to the cost of calculating eigenvalues for large-scale matrices.

**A proxy method: population gradient.** The surveillance network optimization requires historical records to compute the matrices **M** and **P**. However, disease surveillance data are typically sparse in underdeveloped settings, especially for emerging infectious diseases. Moreover, the SA algorithm is computationally expensive and prohibitive for systems with more than a few hundred locations[49]. For large-scale systems or diseases with limited historical records, a practical strategy to design surveillance networks is needed. Given the similarity between the surveillance networks selected by SA and ∇Population, we propose that ∇Population, a metric that is broadly available, can be used to select surveillance sites.

We examined the performance of ∇Population at finer spatial resolution using synthetic influenza outbreaks generated at the county level. Specifically, error-laden observations of ILI+ for 20 outbreaks in the 3108 continental US counties were generated using the mobility-driven metapopulation model (Supplementary Note 7). We then compared the forecasting accuracy of surveillance networks constructed using various, alternate strategies. Specifically, we considered four other heuristic approaches: site selection informed by population coverage, number of commuters, diversity of commuters' residential counties, and random selection. A recent study found that selecting sentinel surveillance sites based on the geographical diversity of patients visiting healthcare facilities performs well for arbovirus disease systems[40]. Here we examined a similar strategy in which counties with more diverse commuters, quantified by the Shannon diversity: $H = -\sum h_i \ln h_i$, where $h_i$ is the fraction of incoming commuters living in county $i$, are preferentially selected. To provide an alternate strategy that avoids geographical clustering, we also included a strategy that randomly selects surveillance sites.

Surveillance networks with $K$ of 5%, 10%, 20% up to 100% of counties were compared. ∇Population outperformed competing strategies (Fig. 4a, Supplementary Fig. 11). Additionally, the marginal reduction of forecast error becomes nominal once 10% of counties are observed. This indicates that observing a small fraction of dynamically central counties is sufficient to generate satisfactory estimates and forecasts for both observed and unobserved locations, and that observing additional sites with potentially larger noise does not necessarily improve forecast accuracy. When we compare results at the state level (Fig. 3d), the

advantage of using ∇Population to design surveillance networks over population and human mobility becomes even more pronounced. This indicates that spatial scale matters in selecting optimal surveillance sites. Indeed, determining the appropriate observational spatial scale that can damp excessive noise while not compromising resolution is a critical, outstanding problem in operational forecasting.

We next compared the overlap of counties selected by ∇Population with those selected by other attributes including local absolute humidity, population, number of commuters, population density, random walk centrality, commuter diversity and random selection (Fig. 4b). With limited overlap, surveillance networks designed using alternate measures differ considerably from the network selected by ∇Population, especially for small numbers of surveillance sites. This comparison indicates that the information conveyed by ∇Population cannot be represented by the other examined metrics.

The competitive performance of ∇Population is explained by its characteristic of avoiding redundant information from clusters of locations: only one population center tends to dominate a cluster of counties. The benefit of avoiding informational redundancy has previously been highlighted[19]. To detail this further, we visualize the surveillance networks composed of 10% of counties as selected by the ∇Population, Population, Commuter, Diversity, and Random approaches (Fig. 4c). Counties selected by the population, commuters and diversity approaches are densely clustered in a few metropolitan areas. In stark contrast, the networks selected by ∇Population are more evenly distributed across the US and are thus more representative of disease activity throughout the country. The randomly selected sites are also distributed across the US; however, many selected counties have small populations with possible large observational noise that could compromise forecasting accuracy.

We quantify geographical clustering using the distribution of distance between nearest neighbors within the surveillance network. The population-, commuter- and diversity-based surveillance networks have on average a closer nearest neighbor (Fig. 4d), indicating a more clustered structure. The networks selected by the random strategy are less clustered, but the distance between nearest neighbors is still slightly lower than that of the population gradient-based networks. For the random strategy, more counties are selected in the eastern and middle US, where counties are more densely distributed. We note that the ∇Population strategy does not merely seek spatial homogeneity; it also reflects the spatial distribution of population: the surveillance sites are denser in areas with more population (Fig. 4c). The SA algorithm also exhibits cluster-avoiding tendencies: during combinatorial optimization, once a location is selected, the chance of selecting an adjacent neighbor is low as the marginal gain diminishes. This mechanism partially explains why the surveillance sites selected by the eigenvalue minimization approach are spread broadly across the US.

We further validated site selection by ∇Population using historical outbreaks for two additional respiratory pathogens: human metapneumovirus (HMPV) and coronavirus (CoV) in 35 US states from 2013–2014 to 2016–2017 (Fig. 5a and Supplementary Note 8). HMPV and CoV are common ILI-causing respiratory viruses, and typically circulate in winter and early spring. In the dataset, their surveillance records are only available in 4 seasons, providing an instance of disease with limited data. Retrospective forecasts for HMPV and CoV outbreaks were generated using surveillance networks composed of different numbers of sentinel sites. Although the signals of HMPV and CoV are noisier than ILI+, due to fewer laboratory tests, the networked forecasting system is still able to predict near-term incidence using partial observations, and the ∇Population site

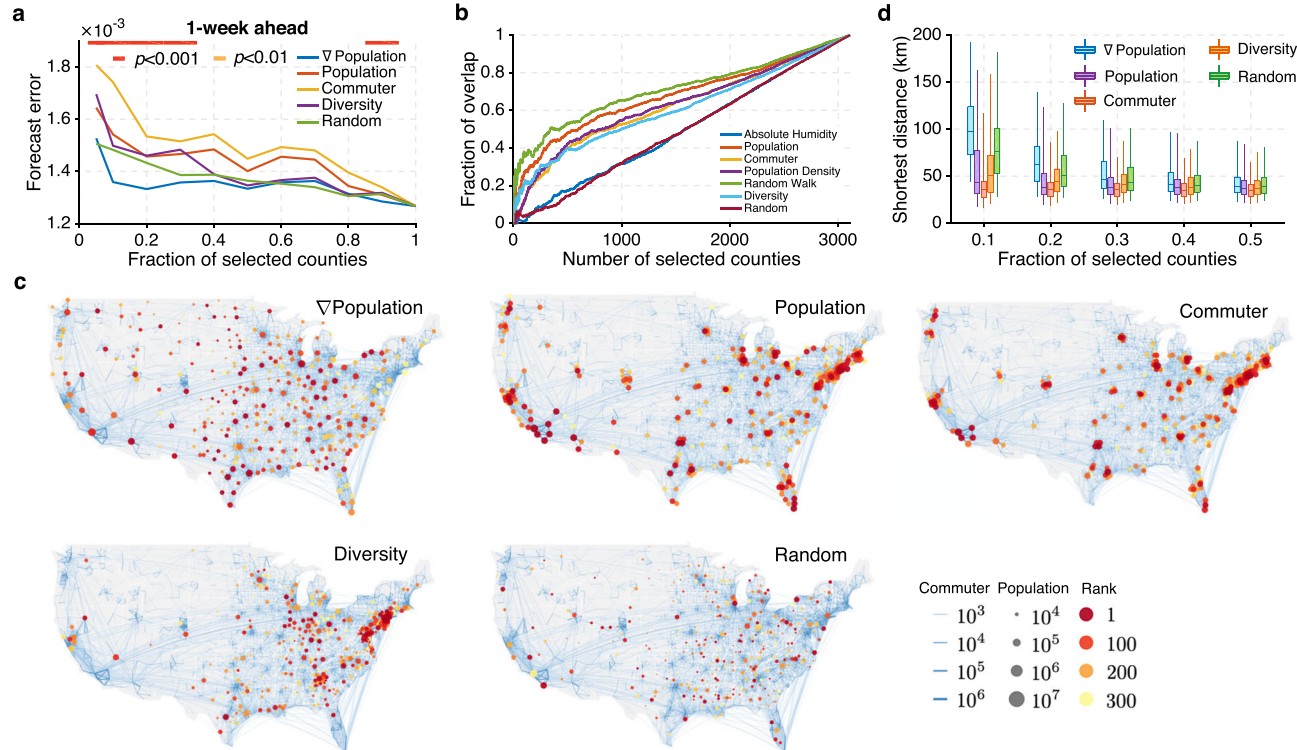

**Fig. 4 Surveillance networks at the county level. a** Forecast error for 1-week ahead ILI+ predictions using surveillance networks consisting of 5%, 10%, 20% up to 100% of all counties. Networks are selected by population gradient (∇Population) (blue line), population (red line), commuter (orange line), diversity of commuters' residential locations (purple line), and random selection (green line). The statistical significance for population gradient outperforming all other methods is reported using the horizontal bars on top (two-sided Wilcoxon signed-rank test; red: $p < 0.001$, orange: $p < 0.01$, none: $p \geq 0.01$). **b** Overlap between the counties selected by ∇Population and those selected by other attributes: absolute humidity (blue line), population (red line), commuter (orange line), population density (purple line), random walk centrality (green line), diversity of commuters' residential locations (light blue line), and random selection (dark red line). **c** Visualization of surveillance networks consisting of 10% of all counties selected by population gradient, population, commuter numbers, commuter diversity and random selection. Blue curves on the map represent county-to-county commuting. Node color indicates the ranking using different methods, and node size reflects the population size. **d** Comparison of the distributions of distance between nearest neighbors within surveillance networks designed using different strategies: ∇Population (blue), population (purple), commuter (red), diversity of commuters' residential locations (orange), and random selection (green). Boxes show the median and interquartile, and whiskers show 95% CI. Distributions were obtained from $n = 310, 621, 932, 1243,$ and $1554$ counties, respectively, corresponding to 10%, 20%, 30%, 40, and 50% of all counties.

selection approach identifies key surveillance locations that support forecasts with lower errors (Fig. 5b–c and Supplementary Fig. 12). The findings demonstrate that forecasting for a range of respiratory viruses is possible in locations without surveillance.

## Discussion

While similar in performance to SA optimization, ∇Population remains a static metric, reflecting only the geographical distribution of population. In contrast, the combinatorial optimization approach using SA accounts for connectivity between locations, observation uncertainty, and evolving model dynamics, and thus more flexibly responds to surveillance practices and outbreak patterns. Nevertheless, should insufficient data (e.g., historical data or estimation of observational error) exist to perform SA optimization, the population gradient method could serve as a reasonable proxy for network site selection. Recent work has revealed the crucial role that urban centers play in incubating and driving influenza transmission[52]; here we identify the significant role metropolises and centers of population play in suppressing uncertainty growth.

As an approximating solution to a combinatorial optimization problem, the optimized surveillance network may have multiple configurations with similar performance[49], i.e., the network constructed using SA is only one of these possible choices. If certain locations are already monitored, such constraint could be

properly incorporated into the optimization problem to find the conditional optimal design for adding more surveillance sites.

Network approaches are increasingly employed in infectious disease modeling, surveillance, and forecasting. In these applications, networked models are usually fitted to real-world observations using computational Bayesian techniques (e.g., Markov Chain Monte Carlo[53], particle filter[54], Kalman filter[55], approximate Bayesian computation[56], etc.). Through this model calibration process, distributions of prior and posterior model states are obtained. This allows the direct quantification of uncertainty propagation when theoretical analysis is intractable and facilitates the generalization of the framework proposed in this study. One possible application would be to assess the value of specific observations and design proactive and adaptive observations (in space and time) in response to an ongoing outbreak.

In the framework used here, important factors affecting influenza outbreaks (e.g., vaccination coverage and effectiveness, mixing patterns within and across age groups, antigenic drift, etc.) were not explicitly represented in the dynamical model. Directly accounting for those factors could potentially further reduce model misspecification and improve the selection of an optimal network. We also only compared the optimization framework with simple location features such as population size and number of commuters. In the future, other more sophisticated strategies for designing surveillance networks could be considered

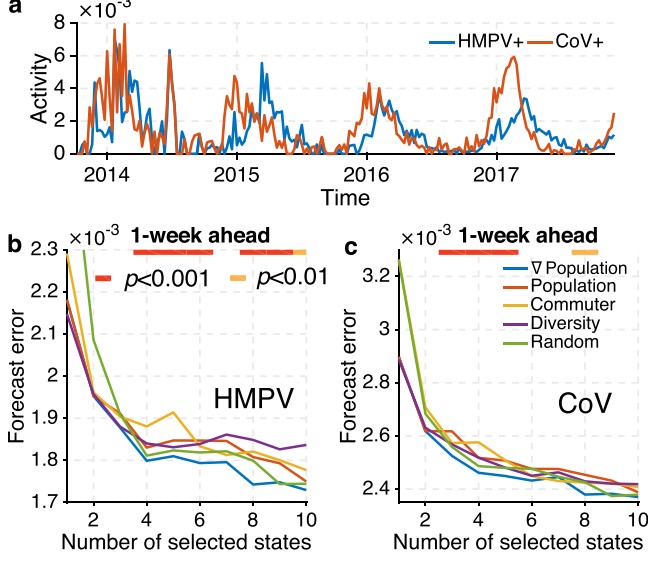

**Fig. 5 Retrospective forecasts for human metapneumovirus (HMPV) and coronavirus (CoV). a** The national HMPV+ rate (blue line) and CoV+ rate (red line) for the 2013–2014 to 2016–2017 seasons from AFHSB data. Forecast errors for 1-week ahead predictions of HMPV and CoV in 35 US states from 2013 to 2017 are reported in (**b**) and (**c**). We focus on surveillance networks consisting of less than 10 states because monitoring more states only provides nominal improvement. The horizontal bar on top indicates the statistical significance for ∇Population (blue line) outperforming methods based on population (red line), number of commuters (orange line), diversity of commuters' residential locations (purple line), and random selection (green line) (two-sided Wilcoxon signed-rank test; red: $p < 0.001$, orange: $p < 0.01$, none: $p \geq 0.01$).

should data and resource availability be sufficient to support proper implementation. Also, the framework only considers the short-term evolution of uncertainty in a linearized approximation. A quantification of longer-term uncertainty propagation in the full nonlinear model would be needed to enhance and optimize the forecast of seasonal targets such as peak timing and peak intensity.

## Methods

**Data description**. We used patient syndromic influenza-like illness (ILI) data and laboratory test results from the US Armed Forces Health Surveillance Branch (AFHSB) to estimate state-level respiratory disease activity (Supplementary Note 1). We focused on the 35 US states in the AFHSB dataset with substantive ILI and test records. For influenza, we used ILI+, defined as the weekly ILI rate among patients seeking medical attention multiplied by the concurrent weekly positivity rate for influenza type A in laboratory testing, to reflect local influenza activity spanning 9 seasons from 2008–2009 to 2016–2017. For HMPV and CoV, laboratory test results are only available for 4 seasons from 2013–2014 to 2016–2017. Similarly, we used ILI multiplied by concurrent positivity rates for these viruses, termed HMPV+ and CoV+ respectively, to estimate disease activity in each state. The ILI visit and laboratory test data were stored in MySQL 8.0 and analyzed in MATLAB 2015b. The use of the deidentified dataset in this study was approved by AFHSB. All relevant ethical regulations were followed.

Local absolute humidity (AH) conditions for each state and county were obtained from North American Land Data Assimilation System data[57]. A daily AH climatology of conditions averaged over a 24-year period from 1979 to 2002 was used. County-to-county commuting data, obtained from the 2009–2013 American Community Surveys, were used to approximate human movement. This dataset, publicly available from the United States Census Bureau website, provides commuting population estimates across all US counties[58]. Given that the survey period (2009–2013) is close to the forecast seasons, we assume the commuting patterns reported in the census survey data are representative of the study period.

**Forecasting framework**. We describe the transmission of respiratory pathogens using a metapopulation SIRS (susceptible-infected-recovered-susceptible) model, in which different locations are connected by human mobility. In practice, detailed information about human movement is not available in real time. To address this

issue, we assume the volume of human movement between two locations is proportional to the average number of commuters between them. Denote $C_j^i$ as the number of commuters living in location $i$ and commuting to work in location $j$. The number of visitors from location $i$ to $j$ is assumed to be $\theta \bar{C}_j^i$, where $\theta$ is an adjustable parameter and $\bar{C}_j^i$ is the average commuters between location $i$ and $j$. The evolution of transmission is then described by

$$\frac{dI_i}{dt} = \frac{\beta_i S_i I_i}{N_i} - \frac{I_i}{D} - \frac{\theta I_i}{N_i} \sum_{j \neq i} \bar{C}_j^i + \theta \sum_{j \neq i} \frac{\bar{C}_i^j I_j}{N_j}, \quad (3)$$

$$\frac{dS_i}{dt} = \frac{N_i - S_i - I_i}{L} - \frac{\beta_i S_i I_i}{N_i} - \frac{\theta S_i}{N_i} \sum_{j \neq i} \bar{C}_j^i + \theta \sum_{j \neq i} \frac{\bar{C}_i^j S_j}{N_j}. \quad (4)$$

Here $N_i$, $S_i$, and $I_i$ are the number of total, susceptible, and infected population in location $i$; $D$ is the average duration of infection; $L$ is the average during of immunity; and $\beta_i$ is the transmission rate in location $i$. The last two terms in the above equations describe the exchange of population due to human movement. For influenza, the transmission rate is modulated by local AH conditions through $\beta_i(t) = [\exp(a \times q_i(t) + \log(R_{0\max} - R_{0\min})) + R_{0\min}]/D$, where $q_i(t)$ is daily specific humidity, a measure of AH. The parameter $a = -180$ is estimated from laboratory experiments of the impact of AH on influenza virus survival. $R_{0\max}$ and $R_{0\min}$ are the maximum and minimum daily basic reproductive numbers inferred during data assimilation. For HMPV and CoV, we assume the transmission rate is constant and identical across locations.

The transmission model is coupled with a data assimilation algorithm to optimize the model state using observed incidence data in real time. Specifically, we used the Ensemble Adjustment Kalman Filter (EAKF)[59] in which the distribution of the model state is represented by an ensemble of state vectors. During data assimilation, this ensemble is iteratively updated so that the model better estimates the underlying unknown truth. The optimized dynamical model is then integrated into the future to generate probabilistic forecasts. Similar model-data assimilation forecast frameworks have been successfully used for forecasting and inference of a variety of infectious diseases[3,60–65]. Details about the system configuration can be found in Supplementary Note 2. The EAKF algorithm was coded in MATLAB 2015b.

**Cross-location uncertainty reduction**. We derived the form of $u_{j \leftarrow i}^I$ and $u_{j \leftarrow i}^S$ analytically using a state-space framework (Supplementary Note 5):

$$u_{j \leftarrow i}^I = \frac{\sigma_{y_i I_j}^2}{\left(R_i + \sigma_{y_i}^2\right) \sigma_{I_j}^2}, \quad u_{j \leftarrow i}^S = \frac{\sigma_{y_i S_j}^2}{\left(R_i + \sigma_{y_i}^2\right) \sigma_{S_j}^2}, \quad (5)$$

where $y_i$ is the prior incidence (i.e., simulated ILI+ rate) in location $i$, $\sigma_{y_i I_j}$ ($\sigma_{y_i S_j}$) is the covariance between the prior incidence in location $i$ and the prior infected (susceptible) population in location $j$, $R_i$ is the OEV of the observation from location $i$, $\sigma_{y_i}^2$ is the variance of the prior incidence in location $i$, and $\sigma_{I_j}^2$ ($\sigma_{S_j}^2$) is the variance of the prior infected (susceptible) population in location $j$. Note that $\sigma_{y_i I_j}$ ($\sigma_{y_i S_j}$) quantifies the dynamical coupling between the observed state variable ($y_i$, simulated ILI+) in location $i$ and the infected (susceptible) population in location $j$. In addition, a more uncertain observation in location $i$ (i.e., a larger $R_i$) leads to a smaller reduction of uncertainty in $I_j$ and $S_j$. In practice, the quantities defining $u_{j \leftarrow i}^I$ and $u_{j \leftarrow i}^S$ in Eq. (5) can be computed numerically using the state-vector ensemble during data assimilation. We validated Eq. (5) in retrospective forecasts of influenza outbreaks over 9 seasons (Supplementary Fig. 5). The actual uncertainty reduction in the state-vector ensemble agrees well with the values calculated using Eq. (5).

**Optimization using simulated annealing**. The configuration vector **p** can be optimized using general iterative optimization algorithms such as simulated annealing (SA)[49]. In SA, the energy function $E(\mathbf{p})$ is defined as $E(\mathbf{p}) = \langle \lambda_1(\mathbf{p}, t, \mathbf{z}) \rangle$. Starting from a random initial configuration that satisfies $\sum_{i=1}^m p_i = K$, at each step $k$, the current configuration vector $\mathbf{p}_k$ is perturbed to $\mathbf{p}_k'$ under constraint of the number of selected locations. This procedure can be realized by swapping the states of a randomly chosen couple of selected and omitted locations. The change in energy, $\Delta E = E(\mathbf{p}_k') - E(\mathbf{p}_k)$, can then be calculated directly from the ensemble of eigenvalues. If $\Delta E < 0$, the perturbation is accepted and the new configuration is used as the starting point for the next step $\mathbf{p}_{k+1} = \mathbf{p}_k'$. Otherwise, the new configuration is only accepted with a probability $P(\Delta E) = \exp(-\Delta E/(\kappa_B T_k))$, where $\kappa_B$ is a constant and $T_k$ is a time-varying parameter called temperature. In implementation, the annealing schedule starts from a high temperature $T_0$, where essentially all perturbations can be accepted, and then gradually cools down to a low temperature with a decreasing probability of accepting worse configurations. The algorithm stops when the number of attempts exceeds a certain threshold value before a new configuration is accepted. The final configuration $\mathbf{p}_\infty$ is the estimated optimal solution to the optimization problem. In our implementation, we used $\kappa_B = 0.1$, an exponentially decreasing temperature $T_k = 0.9997^k$ and a

maximal iteration number of $k_{max} = 30,000$. The stopping threshold was set at 3000.

**Evaluation of retrospective forecasting**. We examined forecast accuracy for 4 short-term targets: 1- to 4-week ahead ILI+ rates. The performance of forecast accuracy is evaluated using two measures: mean absolute error (MAE) and log score. MAE is calculated as the difference between the predicted ensemble mean and the observed ILI+ rate. Log score is defined as the log value of the probability assigned to the interval of width 0.01 centered at the observed ILI+ rate (0.005 on each side)[15–17].

In order to examine whether the SA algorithm statistically significantly outperforms the other three strategies in retrospective forecasting for influenza outbreaks, we performed a Wilcoxon signed-rank test on three pairs of methods: SA-Population, SA-Commuter, and SA-VPopulation. The Wilcoxon signed-rank test is a non-parametric statistical test that compares two paired samples (here, paired MAEs or log scores generated by both examined methods for the same location at the same forecast week) to assess whether their mean-ranks differ[66]. We performed a two-sided test to return a p-value indicating that SA outperforms the other method. We calculated the p-values for the three pairs of comparison (SA-Population, SA-Commuter, and SA-VPopulation) for each of the four targets. The p-values reported in Fig. 3d–e are the maximal p-values among all three tests (i.e., the worst case). The same analysis was performed for forecasting at the county level and for HMPV and CoV.

**Reporting summary**. Further information on research design is available in the Nature Research Reporting Summary linked to this article.

## Data availability

The US commuting data is available at https://www2.census.gov/programs-surveys/demo/tables/metro-micro/2015/commuting-flows-2015/table1.xlsx. The disease surveillance data that support the findings of this study are available from AFHSB but restrictions apply to the availability of these data, which were used under license for the current study, and so are not publicly available. Data are however available from the authors upon reasonable request and with permission of AFHSB. Source data for part of the figures are provided with this paper. Source data are provided with this paper.

## Code availability

The code for the networked forecasting system is deposited in GitHub at https://github.com/SenPei-CU/SurveillanceOptimization.

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

## Acknowledgements

This work was supported by US National Institutes of Health grant GM110748, Defense Advanced Research Projects Agency contract W911NF-16-2-0035, and a gift from the Morris-Singer Foundation.

## Author contributions

S.P. and J.S. designed the research; S.P. and X.T. performed the experiments and analysis; P.L. curated the data; S.P., X.T., P.L., and J.S. interpreted the results and wrote the manuscript.

## Competing interests

J.S. and Columbia University disclose partial ownership of SK Analytics. J.S. discloses consulting for BNI. All other authors declare no competing interests.
