## [Peer Review File · Nature Communications]

Reviewers' Comments:

Reviewer #1:

Remarks to the Author:

In their manuscript, Pei et al. use a framework for selecting a subset of influenza surveillance sites and show the accurate forecasts are possible even when there are gaps in the surveillance system. Despite the growing interest in forecasting infectious disease outbreaks, comparatively little effort has been spent on how to gather data optimally for these forecasting systems. As a result this is both a timely and impactful contribution. I do have a few questions/comments, which I hope the authors find constructive.

1. For many geographic locations and/or diseases, historical data either do not exist or are unreliable. One benefit of using your "alternative" models, i.e., selecting on population and mobility, is that these data tend to be much more common. Interestingly, Scarpino et al. 2017 found that by selecting on the diversity of patients visiting healthcare facilities, where diversity is measured by patient zip code, this was almost as accurate as having historical data on disease. Could you envision a similar heuristic in this system?

Scarpino, S. V., Meyers, L. A., & Johansson, M. A. (2017). Design strategies for efficient arbovirus surveillance. *Emerging infectious diseases*, 23(4), 642.

2 A number of other studies look at optimal data integration. In their models, although often not explicitly stated as subset selection, they are performing some kind of regularization on the number of predictors, which is very nearly subset selection. Therefore, I don't agree with your statement that, "However, despite widespread use of network approaches, no work has explored disease surveillance and forecasting using incomplete data."

However, I think your work is still an important contribution. My recommendation is to cite more of these papers and modify the sentence.

Santillana, M., Nguyen, A. T., Dredze, M., Paul, M. J., Nsoesie, E. O., & Brownstein, J. S. (2015). Combining search, social media, and traditional data sources to improve influenza surveillance. *PLoS computational biology*, 11(10).

Herrera, J. L., Srinivasan, R., Brownstein, J. S., Galvani, A. P., & Meyers, L. A. (2016). Disease surveillance on complex social networks. *PLoS computational biology*, 12(7).

Ertem, Z., Raymond, D., & Meyers, L. A. (2018). Optimal multi-source forecasting of seasonal influenza. *PLoS computational biology*, 14(9), e1006236.

Polgreen P, Chen Z, Segre A, Harris M, Pentella M, et al. (2009) Optimizing influenza sentinel surveillance at the state level. *Am J Epidemiol* 170:

4.) Your method is seemingly a more general version of the approach in Scarpino et al. 2012. If so, you should be able to efficiently solve your objective function with a greedy algorithm. See Das and Kempe 2008. What do the results look like with a greedy algorithm, as compared to simulated annealing? Are you able to use the theorems in Das and Kempe 2008 to compute a theoretical upper-bound on the value of your objective function for any given surveillance system size?

Das A, Kempe D (2008) Algorithms for subset selection in linear regression. *Proceedings of the 40th annual ACM symposium on Theory of computing*. New York: ACM. pp. 45–54.

5.) I would be curious to see how well all of the methods perform against simply selecting subsets at random.

Reviewer #2:

Remarks to the Author:

This submission uses a careful approach to high-quality data to ask important questions.

The methods are complicated, but seem adequately validated by the experiments that the authors do.

The heuristic story is nice, but has a little bit of a straw-person feeling to it. Other than the ∇P , all of the heuristics felt from the beginning like they were going to be over-clustered. Is there any information about what sorts of heuristics people actually use for applications like this one?

The equation following L119 is problematic in its current form. In correspondence, the authors have agreed to use a different form for the revision.

Jonathan Dushoff

Reviewer #1 (Remarks to the Author):

In their manuscript, Pei et al. use a framework for selecting a subset of influenza surveillance sites and show the accurate forecasts are possible even when there are gaps in the surveillance system. Despite the growing interest in forecasting infectious disease outbreaks, comparatively little effort has been spent on how to gather data optimally for these forecasting systems. As a result this is both a timely and impactful contribution. I do have a few questions/comments, which I hope the authors find constructive.

Response: We appreciate the reviewer's constructive suggestions and comments. Please find a point-by-point response to reviewer's questions below.

1. For many geographic locations and/or diseases, historical data either do not exist or are unreliable. One benefit of using your "alternative" models, i.e., selecting on population and mobility, is that these data tend to be much more common. Interestingly, Scarpino et al. 2017 found that by selecting on the diversity of patients visiting healthcare facilities, where diversity is measured by patient zip code, this was almost as accurate as having historical data on disease. Could you envision a similar heuristic in this system?

Scarpino, S. V., Meyers, L. A., & Johansson, M. A. (2017). Design strategies for efficient arbovirus surveillance. *Emerging infectious diseases*, 23(4), 642.

Response: We thank the reviewer for this suggestion. In the revision, we develop a heuristic strategy based on the diversity of commuters' residential locations. Specifically, we examine a strategy that preferentially selects sites with more diverse commuters, quantified by the Shannon diversity: $H = -\sum h_i \ln h_i$. Here h_i is the fraction of incoming commuters living in location i . We apply this heuristic method to influenza surveillance at county level, and HMPV and CoV surveillance at state level. At county level, the networks selected by commuter diversity have broader geographical coverage than population- and commuter-based networks. However, the selected networks still have clusters in several metropolitan areas. In forecasting experiments, the population gradient-based networks generally outperform the commuter diversity-based networks at both county and state levels. We note that this result does not contradict with the conclusion in Scarpino et al. 2017, as the definition of diversity and model scale are different. Please see lines 303-313, 336-352 and Figs. 4-5 in the revised manuscript.

2 A number of other studies look at optimal data integration. In their models, although often not explicitly stated as subset selection, they are performing some kind of regularization on the number of predictors, which is very nearly subset selection. Therefore, I don't agree with you statement that, "However, despite widespread use of network approaches, no work has explored disease surveillance and forecasting using incomplete data."

However, I think your work is still an important contribution. My recommendation is to cite more of these papers and modify the sentence.

Santillana, M., Nguyen, A. T., Dredze, M., Paul, M. J., Nsoesie, E. O., & Brownstein, J. S. (2015). Combining search, social media, and traditional data sources to improve influenza surveillance. *PLoS computational biology*, 11(10).

Herrera, J. L., Srinivasan, R., Brownstein, J. S., Galvani, A. P., & Meyers, L. A. (2016). Disease surveillance on complex social networks. *PLoS computational biology*, 12(7).

Ertem, Z., Raymond, D., & Meyers, L. A. (2018). Optimal multi-source forecasting of seasonal influenza. *PLoS computational biology*, 14(9), e1006236.

Polgreen P, Chen Z, Segre A, Harris M, Pentella M, et al. (2009) Optimizing influenza sentinel surveillance at the state level. *Am J Epidemiol* 170:

Response: Thanks for pointing us to these relevant works. In the revised manuscript, we have modified the statement, and added a paragraph to introduce these studies. Please find changes in lines 52-66 of the revised manuscript.

4.) Your method is seemingly a more general version of the approach in Scarpino et al. 2012. If so, you should be able to efficiently solve your objective function with a greedy algorithm. See Das and Kempe 2008. What do the results look like with a greedy algorithm, as compare to simulated annealing? Are you able to use the theorems in Das and Kempe 2008 to compute a theoretical upper-bound on the value of your objective function for any given surveillance system size?

Das A, Kempe D (2008) Algorithms for subset selection in linear regression. *Proceedings of the 40th annual ACM symposium on Theory of computing*. New York: ACM. pp. 45–54.

Response: We agree that a greedy algorithm is an attractive alternative approach to efficiently solve the optimization problem. At state level, we use a greedy algorithm that sequentially adds locations that lead to the largest marginal reduction of eigenvalue. Specifically, we run a greedy algorithm to select a given number of states and compare the resulting eigenvalue with the one obtained from the SA algorithm (Supplementary Fig. 10). The eigenvalue curves are identical for surveillance systems with less than 15 states and remain similar for larger systems. This comparison indicates that the greedy approach is effective for this 35-state model, and may be applicable to small- and medium-size systems. However, for large systems like the county-level transmission model, the greedy algorithm is still prohibitive due to the cost of computing eigenvalues for large-scale matrices. For instance, to rank all 3000 counties, it is necessary to compute eigenvalues for multiple 3000x3000 matrices for around 4.5M times. In terms of performance guarantee, unfortunately, the objective function in this study (i.e., the eigenvalue function) is not submodular. As a result, a theoretical upper-bound on the eigenvalue is not available.

In fact, we have a separate follow-up study to develop an efficient approximation method that bypasses the computation of eigenvalues. In that study, we approximate eigenvalues using power iteration and show that the approximating objective function is submodular. However, this follow-up study is rather technical and outside the scope of this manuscript. We thus leave this question for a separate future manuscript.

Please find changes in lines 276-290 in the revised manuscript and Supplementary Fig. 10.

5.) I would be curious to see how well all of the methods perform against simply selecting subsets at random.

Response: We added a random selection strategy for influenza surveillance at the county level, and for HMPV and CoV surveillance at the state level. The network selected by the random strategy is less clustered. However, its forecasting performance is still less satisfactory than the population gradient-based networks. The randomly selected sites spread across the US; however, many selected counties have small populations with potentially large observational noise that could compromise forecasting accuracy. In the revised manuscript, we add this additional analysis in lines 303-313, 336-352 and Figs. 4-5.

Reviewer #2 (Remarks to the Author):

This submission uses a careful approach to high-quality data to ask important questions.

The methods are complicated, but seem adequately validated by the experiments that the authors do.

Response: We appreciate reviewer's efforts in evaluating our manuscript.

The heuristic story is nice, but has a little bit of a straw-person feeling to it. Other than the ∇P , all of the heuristics felt from the beginning like they were going to be over-clustered. Is there any information about what sorts of heuristics people actually use for applications like this one?

Response: We thank reviewer for this suggestion. The population-based selection method was proposed in Polgreen et al. *Am. J. Epidemiol.* (2009), which aims to maximize the population coverage of a surveillance system. In the revised manuscript, we have tested two additional heuristic strategies. 1) We preferentially select counties with more diverse commuters, quantified by the Shannon diversity: $H = -\sum h_i \ln h_i$, where h_i is the fraction of incoming commuters living in county i . This strategy was developed in Scarpino et al. *Emerg. Infect. Dis.* (2017) to optimize arbovirus surveillance. 2) We randomly select surveillance sites to avoid geographical clustering of sentinels. We apply those two heuristics to influenza surveillance at county level, and HMPV and CoV surveillance at state level. Results indicate that the performance of the additional methods is in general less satisfactory than the population gradient approach. Please find more details in lines 303-313, 336-352 and Figs. 4-5 of the revised manuscript.

The equation following L119 is problematic in its current form. In correspondence, the authors have agreed to use a different form for the revision.

Response: We have updated the equation and re-run SA optimization for influenza surveillance at the state level. As expected, for a given number of observation locations, the set selected of states remains unchanged. We thank the reviewer for pointing out this issue.

Reviewers' Comments:

Reviewer #2:

Remarks to the Author:

The revision appropriately addresses my concerns.

Reviewer #2 (Remarks to the Author):

The revision appropriately addresses my concerns. The authors should consider whether they want to modify or eliminate their original dismissal of greedy algorithms (L194) in light of their new text.

Response: We appreciate reviewer's efforts in evaluating our manuscript. In the revision, we have removed the statement on greedy algorithms (though the intention of this statement was to stress that the analytical form of gradient approaches is intractable). Please see the change in line 194 in the revised manuscript.